# ISIDOG Consensus Guidelines on COVID-19 Vaccination for Women before, during and after Pregnancy

**DOI:** 10.3390/jcm10132902

**Published:** 2021-06-29

**Authors:** Gilbert G. G. Donders, Svitrigaile Grinceviciene, Kai Haldre, Risa Lonnee-Hoffmann, Francesca Donders, Aristotelis Tsiakalos, Albert Adriaanse, José Martinez de Oliveira, Kevin Ault, Werner Mendling

**Affiliations:** 1Femicare VZW Clinical Research for Women, 3300 Tienen, Belgium; francesca.donders@gmail.com; 2Department Obstetrics and Gynecology, University Hospital Antwerp, 2650 Edegem, Belgium; 3President International Society Infectious Diseases (ISIDOG), 3300 Tienen, Belgium; 4Department Biothemodynamics and Drug Design, Institute of Biotechnology and Life Sciences Center, Vilnius University, 01513 Vilnius, Lithuania; svitrigaile.grinceviciene@bti.vu.lt; 5East Tallin Central Hospital Women’s Clinic, 10138 Tallin, Estonia; kai.haldre@telenet.be; 6Department Gynecology, Hospital St Olav, 7030 Trondheim, Norway; risa.lonnee-hoffmann@stolav.no; 7LETO-Obstetrician Gynecological & Surgical Center, Department Obstetrics and Gynecology, 11525 Athens, Greece; atsiakalos@gmail.com; 8Medisch Centrum Alkmaar, Department Obstetrics and Gynecology, 1814 Alkmaar, The Netherlands; a.adriaanse3@kpnplanet.nl; 9Department Obstetrics and Gynecology, Universidade da Beira Interior, 6201-001 Covilha, Portugal; jmo@fcsaude.ubi.pt; 10Department Obstetrics and Gynecology, University of Kansas Medical Center, Kansas City, KS 66160, USA; kault2@kumc.edu; 11German Center for Infections in Obstetrics and Gynceology, Department Obstetrics and Gynecology, 42283 Wupperthal, Germany; w.mendling@t-online.de

**Keywords:** vaccine, pregnancy complication, SARS-CoV-2, COVID-19, maternal complications, pandemic, prevention, safety

## Abstract

Introduction. Sars-CoV-2 infection poses particular problems in pregnancy, as the infection more frequently causes severe complications than in unaffected pregnant women or nonpregnant women with SARS-CoV-2 infection. Now that vaccination is available and rapidly being implemented worldwide, the question arises whether pregnant women should be vaccinated, and if so, whether they should receive priority. Methods. Available scientific data and available guidelines about vaccination against SARS-CoV-2 were collected by the Guideline Committee of the International Society of Infectious Diseases in Obstetrics and Gynecology (ISIDOG) and were analyzed, discussed and summarized as guidelines for healthcare workers caring for pregnant women. Concluding statements were graded according to the Oxford evidence-based medicine grading system. Results. There is evidence to consider pregnancy as a risk factor for serious complications of COVID-19 infection, even in the absence of additional risk factors, such as hypertension, diabetes and obesity which increase these risks even more in pregnancy. Currently available data slightly favor mRNA-based vaccines above vector-based vaccines during pregnancy and breastfeeding, until more safety data become available. Conclusion. ISIDOG advises policy makers and societies to prioritize pregnant women to receive vaccination against SARS-CoV-2 and favor the mRNA vaccines until further safety information becomes available.

## 1. Introduction

Soon after the start of the Sars-Cov-2 pandemic, it became clear that the discussion regarding prioritization of pregnant women’s vaccination deserved special attention. Indeed, first of all, pregnant women, or women intending to become pregnant, belong to a gender and age group more likely to be employed in healthcare settings, where the risks of acquiring the infection is higher as well as the risk of, once infected, further spreading it to sick, vulnerable people being taken care of in hospitals, rehab centers and nursing homes. In addition, in general, the theoretical risk of transmitting the infection to the offspring during pregnancy, delivery or breast feeding requires surveillance, as well as any potential danger of harming the course of the pregnancy by complications such as fetal anomalies, preterm birth, thrombosis, pre-eclampsia, etc. Finally, being pregnant may influence the course of the COVID-19 disease, due to altered immunity, and ventilation restrictions of the maternal lungs due to limited diaphragm movement as a result of an increasing uterine volume. At the top of the first wave of COVID-19 infection in Europe, in April 2020, we issued International Society of Infectious Disease of Obstetrics and Gynecology (ISIDOG) recommendations on the prevention and treatment issues of Sars-Cov-2 infections in pregnancy, based on the available published data at that moment [1]. Already at that point, it was clear that direct transmission of Sars-Cov-2 to the fetus did not appear to be major concern but rather to prevent or handle the more severe course of the disease during pregnancy, once infected. More premature babies were born due to Caesarian sections and induced labors in an effort to provide more efficient, sometimes lifesaving, oxygen therapy to the pregnant woman.

Sars-Cov-2 vaccines are now being produced by several companies and are gradually approved by the competent authorities, such as FDA (U.S. Food and Drugs administration, USA) and EMA (European Medicines Agency, Europe). How to prioritize provision of a vaccine that aims to cover the entire population of the globe, however, is an enormous task, given the relative shortage of vaccines and lack of means for effective delivery to the people. Indeed, there are the logistic problems of production and delivery of sufficient vaccines in a timely manner by the manufacturers, as well as privacy regulations, age, logistic restriction (e.g., distance), transport requirements (e.g., frozen at −70 or −20 °C), safety issues, side-effect profiles, etc. Especially for pregnant women, the latter points are specifically important issues that need to be addressed before general recommendations can be made.

The guideline commission of the International Society of Infections in Obstetrics and Gynecology (ISIDOG) organized an expert collaboration in order to reach consensus about the recommendations concerning vaccination of women who intend to be, are or recently were pregnant, taking into account the special questions related to these groups, the types of vaccines and the specific risks and considerations. While aiming continuously to expand the participation of countries, ISIDOG currently represents all European countries except Ireland, Iceland, Slovenia, Bosnia-Herzegovinia and Albania, and additionally some other states such as the USA, China, Brazil and the Republic of Congo are represented.

## 2. Methods

This work was performed by a team of leading expert obstetricians to guide fellow healthcare workers in the management of the pregnant women at risk of Sars-CoV-2 infection. Initially, the questions that were required to be answered were defined: [1] Does COVID-19 constitute a high-risk situation during pregnancy that calls for systematic Sars-Cov-2 vaccination during pregnancy?; [2] Can Sars-Cov-2 vaccination be disadvantageous for mother or fetus if given during pregnancy?; [3] Is the type of vaccine used for pregnant women of importance?; [4] Is timing of vaccination in pregnancy of importance?; and [5] Do responses to the above questions also apply for women intending to become pregnant or to lactating women? The literature search in PubMed using ‘Sars-Cov-2′ or ‘COVID-19′ and ‘pregnancy’ or ‘lactation’ and ‘Sars-Cov-2′ or ‘COVID-19′ and ‘vaccine’ or ‘vaccination’ identified, respectively, 2214 and 1424 references, from which information was collected. Of these, 16 papers were found of sufficient quality and novelty to base the consensus guidelines on. We aimed for consensus based on these literature data by all guideline group members. The consensus guidelines we produced were finally graded according the Oxford Centre for Evidence-Based Medicine’s grading system (accessed on 25 March 2019: https://www.cebm.ox.ac.uk/resources/levels-of-evidence/oxford-centre-for-evidence-basedmedicine-levels-of-evidence-march-2009).

## 3. Results and Conclusions

### 3.1. COVID-19 Infection during Pregnancy

#### 3.1.1. Maternal Effects of COVID-19

Although there is no proof the risk to contract Sars-Cov-2 infection is facilitated by pregnancy, the altered immune system, alongside other physiologic changes in pregnancy [2], causes more sensitivity for infection of other RNA viruses such as SARS, Ebola and Marburg virus, warranting for caution [1].

Symptoms of COVID-19 infection are similar in pregnant as compared to nonpregnant women, except for fever, myalgia and sore throat, which are more frequent during pregnancy [3,4]. Like nonpregnant women, the majority of women in pregnancy are also without symptoms [5].

Thromboembolic risk. Pregnant patients with mild vs. severe COVID-19 have a 0.2–6% increased risk, respectively, of thromboembolism with much higher D-dimer levels than uninfected pregnant women. [5,6] Therefore, we advised in our previous ISIDOG guidelines for anticoagulant therapy for pregnant COVID-19 patients admitted into the hospital [1]. An overview of international recommendations concerning thromboprophylaxis of COVID-19 in pregnancy can also be found in a recent review [7].

Since the start of the epidemic, many studies have been launched to assess the impact of Sars-Cov-2 on women during pregnancy. In a study from Centers for Disease Control and Prevention (CDC), Atlanta, data on pregnancy status were present on 461.825 omen with PCR proven Sars-Covid-2 infection. Of the symptomatic women, 434 (5.7%) were pregnant and followed. After adjusting for age, underlying medical conditions and ethnicity, the risk of dying due to Sars-Cov-2 was increased by 70% [8]. In addition, admissions at intensive care units (ICU, 10.5 vs. 3.0 per 1000 cases (adjusted relative risk (aRR) = 3.0, 95% CI 2.6–3.4)), risk of mechanical respiratory ventilation (2.9 vs. 1.1. per 1000 cases (aRR 2.9, 95% CI 2.2–3.8), necessity of heart–lung machine (ECMO, 0.7 vs. 0.3 per 1000 cases (aRR 2.4 95% CI 1.5–4.0) and risk of death (1.5 vs. 1.2 per 1000 cases (aRR 1.7 95% CI 1.2–2.4) due to Sars-Cov-2 infection are increased in pregnant compared to nonpregnant women. As a result, CDC considers the state of pregnancy as a risk factor for increased severity of COVID-19 infection, indicating that pregnant women are a priority group for timely COVID-19 vaccination. (accessed on 21 February 2021: https://www.cdc.gov/coronavirus/2019-ncov/need-extra-precautions/pregnancy-breastfeeding.html).

In 77 eligible papers, Allotey et al. systematically reviewed the risks of Sars-Cov-2 infection on pregnant women. These data confirmed a higher risk of ICU admission (odds ratio (OR) 1.6) and mechanical ventilation (OR 1.9) when infected by SARS-CoV-2 during pregnancy [3]. Moreover, they revealed additional factors that strongly increased the risk of these serious complications: age above 35 years (OR 1.8; 95%CI 1.3–2.6) BMI greater than 30 (OR 2.4; 95%CI 1.7–3.4), hypertension (OR 2.0; 95%CI 1.1–3.5), diabetes mellitus (OR 2.5; 95%CI 1.3–4.8) and pre-eclampsia (OR 6.5; 95%CI 1.1–36.2) all increased the risks significantly.

When extrapolating these data to other parts of the world, it has to be taken into account that some characteristics may differ between populations, especially when looking at absolute numbers. In the US, insufficient or lacking medical insurance may cause a delay in seeking medical and prenatal care. In the US, obesity is more frequent, large ethnic groups have an inherent increased risk of pregnancy complications (e.g., Afro-Americans) and COVID-19 infection is more frequent among certain ethnic groups (Latinos) than in Europe. In 2017, before the Sars-Cov-2 infection was noticed, maternal mortality was almost 5 times higher (19/100,000 births) in US compared to most European countries, such as Belgium (5/100,000) (life born neonates in Belgium) (accessed on 29 March 2021: http://documents1.worldbank.org/../Trends-in-maternal-mortality-2000-to-2017-Estimates-by-WHO-UNICEF-UNFPA-World-Bank-Group-and-the-United-Nations-Population-Division.pdf).

Due to the increased risks of severe complications of COVID-19 in pregnancy and the increased difficulty to provide intensive care treatment, the ISIDOG guidelines advocated to protect pregnant midwifes and other healthcare personnel from the work environment with substantial risk of COVID-19 infection. Further, we plead for considering pregnancy a high-risk factor and therefore as a priority group for vaccination [1].

**Conclusion** **1.**
*Infection risk and symptoms of uncomplicated Sars-Cov-2 are not substantially different in pregnant vs. nonpregnant women (Grade A).*


**Conclusion** **2.**
*In some cases, the course and severity of Sars-Cov-2 infection is worsened by pregnancy, especially in combination with additional risk factors such as increased maternal age, pre-eclampsia, obesity, diabetes and hypertension (Grade A). Therefore, pregnancy should be considered as a priority for vaccination.*


#### 3.1.2. Neonatal Outcome

The main concern for pregnancy outcome in connection with SARS-CoV-2 infection is preterm delivery. Most studies confirm an increased rate of preterm deliveries in SARS-CoV-2-positive pregnant women, mostly because obstetricians choose for induction of labor or primary C-section in order to improve the oxygenation and care of severely ill mothers. For other pregnancy complications, the risk after Sars-Cov-2 infection is very comparable to that of the normal pregnant population [4]. In a population-based cohort study in UK, gestation ended preterm in 26% of women, of which 60% were due to maternal or fetal compromise [9]. In his meta-analysis, Allotey also confirmed a three times higher incidence of preterm births and increased neonatal admissions, mainly caused by iatrogenic interventions to terminate pregnancy [3].

Another matter of concern is the 2–3-fold increased rates of perinatal mortality (both stillbirth and neonatal death) have been reported from two prospective cohort studies in the UK [10] and Italy [11], where 11.7 and 10 per 1000 perinatal deaths (PND) occurred in women with COVID-19, respectively, compared with a national average of 4 per 1000 in both nations. This was confirmed by a systematic review of Khalil et al., where a mean PND rate of 7.8/1000 was noted. [12]. The fetal fatality is thought to be linked to the increased number of intervillous thromboses and decreased placental perfusion in pregnant women with COVID-19 infection [12,13].

The risk of a neonate to be infected at birth is very low or negligible. The infrequent occurrence of transmission might be attributed to the minimal expression of ACE-2 receptor and TMPRSS2 in the placenta [14]. Some case studies suggested transmission could have occurred, resulting in positive placental or umbilical cord Sars-Cov-2 PCR testing, but larger scale investigations; such a transmission has not been confirmed yet. A positive Sars-Cov-2 PCR was shown in about 3% nasopharyngeal samples and 9.7% of rectal swabs of neonates born to COVID-19 positive mothers, but urine and amniotic fluid tests were always negative, suggesting that postnatal, rather than transplacental, transmission may have occurred [15]. In a meta-analysis discussing 176 neonatal COVID-19 infections, the decision not to separate mother and child resulted in 5-fold increase in a neonatal SARS-CoV-2 infection, emphasizing the importance of postpartum transmission [16].

**Conclusion** **3.**
*The increased risk to be born preterm due to iatrogenic intervention, as well as a possibly increased perinatal mortality for babies born to women with COVID-19 infection, supports the need for vaccination during or before pregnancy (Grade A).*


**Conclusion** **4.**
*Direct transplacental transmission of Sars-Cov-2 does not seem to constitute a major argument in favor of Sars-Cov-2 vaccination (Grade C).*


### 3.2. COVID-19 Vaccine Safety in Pregnancy

#### 3.2.1. Need for Studies in Pregnancy

Any intervention in pregnancy, including vaccinations, especially new ones, requires special careful attention to ensure safety for pregnant women and children. In pregnancy, immune reactions against viral infections are usually less intense but can also be hyperacute. This makes it difficult to transpose general data to a pregnant population. With new drugs and vaccinations, no studies on pregnant populations are commonly available. This results in the safety principle that any new vaccine is not for routine use in pregnancy until evidence of its efficacy and safety is proven. On the other hand, this poses the problem that pregnant women may be excluded from the benefits of important and even lifesaving vaccines, as they are systematically excluded from studies dealing with new vaccines.

Vaccines using a viral vector. Both the Astra-Zeneca and the J&J/Janssen COVID-19 vaccines are based on a (adeno-)virus vector to deliver important instructions to our cells. These vaccine types have been tested substantially, both on animals and humans, without negative effects, and are used on a large scale. Similar viral vector vaccines have been given to pregnant people in all trimesters of pregnancy, e.g., in a large-scale Ebola vaccination trial. In these trials, no adverse pregnancy-related outcomes were reported. Formal studies on the effects of these vaccines in pregnancy are lacking (accessed on 1 April 2021: https://www.ema.europa.eu/en/documents/product-information/zabdeno-epar-product-information_fr.pdf). Nevertheless, as rare side effects of venous sinus thrombosis of the brain and splanchnic thromboses only seem to occur in adenovectored vaccines, any further decisions to use these vaccines in pregnancy should perform risk–benefit analysis based on best available data on potential thromboembolitic AE and in light of exposure risk to COVID-19 based on the epidemiological situation.

From vaccines based on the messenger-RNA, such as the COVID-19 vaccines manufactured by Pfizer-BionTech and Moderna, no official study data were published so far. Animal studies did not reveal any impact on the offspring of rat exposed to the vaccine. As mRNA vaccines do not contain the live virus that causes COVID-19, it cannot cause COVID-19 infection. Additionally, mRNA vaccines do not interact with a person’s DNA or cause genetic changes because the mRNA does not enter the nucleus of the cell. COVID-19 vaccine manufacturing companies announced their intention to perform studies on pregnant subjects in order to confirm the safety of the vaccine for mother and child. Pfizer started as the first company with a phase 2 and 3 study with 4000 pregnant women to ensure safety during the second and third trimesters of pregnancy, as well as to monitor potential effects on the newborns.

In the US, the V(accination)-Safe program allows participants to voluntarily enter their personal information on a website. The participants may receive follow-up text messages and phone calls from the CDC asking for additional information at various times after vaccination. This information is collected and connected to information from the Vaccine Adverse Event Reporting System (VAERS), where healthcare workers, patients and other people can report adverse events on a standardized form. In their latest communication, 55,220,364 reports had been received from people who received at least one dose of the Pfizer-BioNTech or Moderna vaccine, including 30,494 pregnant women, of whom 16,039 had received the Pfizer-BioNTech and 14,455 women the Moderna vaccine. (Medscape–accessed on 10 March 2021, https://www.medscape.com/viewarticle/947211). The rates of side effects and complications in vaccinated women do not appear significantly different from those of unvaccinated pregnant women. The additional follow up of 1815 pregnant women, of which 275 already completed pregnancy, revealed similar or lower rates of miscarriages, preterm births, stillbirths, hypertensive disorders, diabetes, growth restriction and perinatal mortality than what were expected from population-based estimates. Based on these surveillance data, CDC suggests that vaccination with the Pfizer-BioNTech and Moderna vaccines for COVID-19 is safe for pregnant women.

#### 3.2.2. Post Vaccination Anaphylaxis

A specific adverse event that can be particularly serious is the risk of postvaccination anaphylaxis.

Women seem to be more sensitive to this complication. In the VAERS system, reporting on almost 10 million women having received Pfizer-BioNTech and over 7.5 million having received Moderna vaccines, 47 and 19 women reported an anaphylactic reaction, respectively, accounting for a rate of 2.5–4.7 per million people vaccinated [17]. Strikingly, 94% and 100% of the anaphylactic reactions in both vaccinated groups, with Pfizer-BioNTech and Moderna, respectively, occurred in women. Of them, respectively 77% and 88% had allergic reactions before, of which 25% and 34% were anaphylactic. An update on these numbers can be found at https://www.nejm.org/doi/full/10.1056/NEJMoa2104983 (accessed on 2 May 2021).

The potential risk of thromboembolic events raised concerns after being noted in some vaccinated women after having received COVID-19 vaccination, especially of the Astra-Zeneca type. The majority of such postvaccine events were encountered in women, below 50 years of age, and some with fatal outcome. This has led some countries in Europe to arrest use of this type of vaccine, while others continued their vaccination policy. As the array of clinical pictures was very diverse (generalized disseminated thrombocytopenia as well as localized thrombo-embolic phenomena in bowel or brain), it took some time to realize this rare condition is related to the use of such vector vaccines. Furthermore, the risk of having serious complications after COVID-19 infection in pregnancy, including thromboembolism, is much greater, leading EMA to the decision (18 March 2021) that the benefits of vaccination outweigh the potential risks. However, an increased surveillance is warranted. During pregnancy, being a hypercoagulative condition in itself, this applies even more. However, as the potential side effects are very rare, and even less likely to occur after a second dose if the first dose went uneventfully, we advise to continue the vaccination schedule according to plan if a person becomes pregnant between the first and the second dose of a vector vaccine.

**Conclusion** **5.**
*Formal studies in pregnancy are lacking. However, follow-up of over 30,000 pregnant women having received m-RNA vaccines reported no increase in side effects or complications (Grade B).*


**Conclusion** **6.**
*As women in general, and particularly during pregnancy, are more vulnerable to anaphylactic reactions and thromboembolic events, these complications require extra surveillance (Grade B). It should be clear, however, that there are no signals that the extremely rare side effects linked to the adenovirus vectored vaccines, are more frequent during pregnancy. In addition, the anaphylactic reactions are rather linked to an history of anaphylaxis than to gestation itself.*


### 3.3. Recommendations of Vaccination during, before and after Pregnancy

Pregnant women and their unborn offspring appear to be a particular risk group for COVID-19 complications. On the other hand, potential complications such as thromboembolism and anaphylaxis may pose more risks for pregnant women than for nonpregnant women or men. Formal studies in pregnant women are currently lacking, but secondary subgroup analysis in large observational population-based studies indicate that the currently available vaccines are safe and do not cause more side effects during pregnancy. Based on currently available knowledge, we recommend that pregnant women should receive priority vaccination. Guidelines are dynamic and should be adapted according to new information.

#### Recommendations of Countries and Societies

The American College of Obstetrics and Gynecology, the Canadian Society of Obstetrics and Gynaecologists and the German Society for Gynecology and Obstetrics all favor pregnant women having the option to be vaccinated and state that “COVID-19 vaccines should not be withheld from pregnant individuals, nor from breastfeeding women. (ACOG Clinical, Vaccinating Pregnant and Lactating Patients against COVID-19, 15 December 2020, SOGC Déclaration de la SOGC sur la vaccination contre la COVID-19 pendant la grossesse, 4 January 2021), and advice which was also supported by others [18,19]. In UK, the RCOG (Royal College of Ob/Gyn) guidelines have recently been adapted in favor of using COVID-19 vaccines in pregnancy. They follow the latest advice from the British Joint Committee on Vaccination and Immunisation (JCVI), namely that COVID-19 vaccines should be considered for pregnant women when their risk of exposure to the virus is high and cannot be avoided, or if the woman has underlying conditions that place her at high risk of complications from COVID-19. In addition, women should not stop breastfeeding in order to be vaccinated against COVID-19, and women trying to become pregnant do not need to avoid pregnancy after vaccination, while there is no evidence to suggest that COVID-19 vaccines affect fertility. In addition, the World Health Organisation (WHO) holds a position ‘in between’, proposing to give the vaccines only to pregnant women with an additional risk factor or who have a high risk of being exposed to the Sars-Cov-2 virus (e.g., healthcare workers), but allegedly, their advice recently was adjusted toward the promotion of general vaccination for all pregnant women. (accessed on 29 January 2021. https://www.fox32chicago.com/news/who-changes-covid-19-vaccine-recommendation-for-pregnant-women/ and https://www.who.int/publications/i/item/WHO-2019-nCoV-vaccines-SAGE_recommendation-AZD1222-2021.1). Even more pronounced, the Flemish Society of Obstetricians and Gynaecologiscts (VVOG) and the Israeli Society for OB/GYN favor vaccination of pregnant women as a priority group. In the Netherlands, on the contrary, the opinion of the NVOG is that there is still not enough evidence to support routine vaccination in pregnancy, but if benefits outweigh the risk, such as in women with co-morbidities, the vaccine should be proposed (NVOG) (accessed on 18 February 2021, https://www.nvog.nl/actueel/standpunt-vaccinatie-tegen-covid-19-rondom-zwangerschap-en-kraambed/). In Belgium, on 8 April 2021, the High Medical Council, advising the Ministry of Health, accepted a motion that sufficient evidence permits one to see pregnancy as a high risk of severe COVID-19 complications, irrespective of additional risk factors, justifying this group to be vaccinated with priority.

**Conclusion** **7.**
*Most European and North American countries are in favor of giving pregnant women the advantages of vaccination, which is considered more beneficial than getting COVID-19 in pregnancy (Grade C). This is irrespective of the presence of additional risk factors that might be present.*


### 3.4. Which Vaccine to Choose in Pregnancy

Other than against COVID-19, there is more experience with several vector-based vaccines in pregnancy. However, concerning COVID-19 vaccines, currently, we have the most experience with the mRNA vaccines. Both types have high-safety profiles outside of as well during pregnancy, but recent unexpected adverse effects after Astra-Zeneca vaccines (Vaxzevria^®^), such as thromboembolic phenomena, currently under investigation, indicate restrictive use in pregnancy and postpartum, because pregnancy constitutes a higher risk of coagulopathies. For the Janssen-COVID-19 vaccine^®^, which is also vector based, very little information is available in pregnancy, but J&J/Janssen’s Ebola vaccine, based on a similar structure, has shown to be very safe in pregnancy. However, given the small sample size of these studies as compared to the COVID-19 vaccines currently applied worldwide, it may well be that the rare thromboembolic events such as venous sinus thrombosis of the Ebola vaccines are still ‘under the radar’.

**Conclusion** **8.**
*Given the uncertainty on potential increased risks of thromboembolic complications in young women after vector-based vaccines such as Astra-Zeneca’s and Janssens’, we currently advocate to use mRNA vaccines in pregnancy, where available, until more safety data become available (Grade D).*


### 3.5. Timing of Vaccination

A prospective study supported by the U.S. National Institutes of Health describes 84 pregnant, 31 lactating and 16 nonpregnant women receiving the mRNA vaccine. Pregnant women receiving the vaccine developed equivalent antibody titers as nonpregnant women (two doses being better than one), and titers were significantly higher than those induced by natural SARS-Cov-2 infection during pregnancy. Secondly, antibodies are proven to cross the placenta (umbilical cord samples) and breast milk, suggesting possibility for passive immunization of the newborn after vaccination of the mother. Infected newborns have a symptomatic COVID-19 infection in 50% of cases; however, severe complications of COVID-19 in newborns are rare [16].

Since transmission of antibodies generally occurs in the third trimester and vaccine titers did not differ based on the trimester in which the vaccine is administered, administration of the vaccine before the third trimester is desirable to achieve the highest maternal titers and passive immunization of the newborn before birth. Vaccination in the first trimester (before 16 weeks) however remains unclear due to limited data on teratogenicity.

**Conclusion** **9.**
*If pregnant during vaccination, antibodies can cross the placenta and possibility for passive immunization of the newborn (Grade A). Preferable time for vaccination in pregnancy is the second trimester, but if women are encountered first during first or third trimester, there is no reason to refrain from vaccination, as the benefits for the mother are likely to outweigh the risks of vaccination or a possible COVID-19 infection during pregnancy (Grade B).*


### 3.6. Women Wanting to Become Pregnant, Postpartum and Breastfeeding Women

Women who intend to become pregnant are at no excess risk of having negative effects of being vaccinated. The effects of the vaccine last a maximum of three weeks for building an immune response, and immunity-related effects, including disseminated intravascular coagulation, are not to be expected beyond that period. Women intending to become pregnant are advised to finish their vaccination schedule before pregnancy in order to have maximal protection for both mother and neonate during pregnancy [20]. Women getting pregnant after the first dose should not be worried, as there is no reason to believe the vaccine causes negative effects in pregnancy, even when provided in the first trimester. According to ESHRE’s guidelines, women having received one or two doses of the vaccine do not need to wait longer than a few days to get pregnant. However, couples engaging in fertility enhancing procedures (assisted reproductive technologies, ART), are advised to wait for 2 months to allow antibody development, as the effect on oocytes, sperm and embryo implantation is still insufficiently examined (accessed on 12 January 2021, https://www.eshre.eu/Europe/Position-statements/COVID19/vaccination). In addition, when a severe immune reaction has taken place in response to the vaccine, a 2 months delay with ART is advised.

**Conclusion** **10.**
*Women who intend to become pregnant can safely be vaccinated and get pregnant soon after vaccination (Grade C). When extracorporeal fertilization is necessary (ART) or when a severe immune response occurred subsequent to vaccination, a 2 months delay is advised, but this is not based on strong evidence (Grade D).*


Postpartum vaccination. There is ongoing discussion about the need to separate mother and child in order to avoid infection of the newborn after COVID-19 positive infection of the mother. A meta-analysis studying 261 neonates born to mothers with COVID-19 infection showed that 10% of those neonates tested positive with PCR tests for Sars-Cov-2. Of these, 20% had light dyspnea or fever, but no serious complications [21]. Placenta, cord blood and vaginal secretions remained negative for SARS-CoV-2 with PCR tests. As vaccination drastically reduces the risk of transmission and there is doubt about the risks of transmission to the neonate after maternal COVID-19 infection, postpartum vaccination can be safely considered.

**Conclusion** **11.**
*As there are conflicting data about the infection of the neonate, vaccination is expected to reduce the risk of mother–child transmission of Sars-Cov-2. Based on current knowledge, postpartum vaccination appears safe (Grade D).*


Breastfeeding. In general, vaccines are not contraindicated in breastfeeding women. As COVID-19 vaccines, especially those based on mRNA technology, cannot transmit virus into the breastmilk, and further protection against infection also protects the neonate against Sars-Cov-2 infection, vaccination should be promoted for breastfeeding women [19]. Other arguments are that SARS-CoV-2 antibodies are passed over by breast milk [22,23,24], especially after vaccination [25] and that most neonatal infections are passed on by caregivers who are not the parents, according to an Italian study [26].

**Conclusion** **12.**
*Breastfeeding women should be encouraged to be vaccinated (Grade C).*


### 3.7. Medico-Legal Issues

Maternal vaccination implies accepting a preventive action that affects at least two people (pregnant mother and unborn child), and therefore medico-legal questions are always implied. In routine care, outside pregnancy, some vaccinations such as polio vaccinations are obligatory in many countries, and others, such yellow fever, are also mandatory if one wants to travel to endemic areas. In pregnancy, other vaccines, such as pertussis and flu (during the winter season), are highly recommended to be given during the second trimester of pregnancy. For COVID-19, specifically, a significantly increased risk of becoming seriously ill or dying as a result of Sars-Cov-2 infection made several health advisers and policy makers decide to offer or even promote Sars-Cov-2 vaccination to women during pregnancy, to protect themselves as well as their babies against such severe complications. As experience is still limited and potential side effects of such vaccines during pregnancy can still surface at a later point, it should be clear that this decision is voluntary, and it is the mother and her partner who finally decide whether, after informed consent, the vaccination is accepted or not.

## 4. Summary

Accumulating data show that COVID-19 infection: has a higher risk of causing severe disease, requiring more admissions to intensive care units; increases the need for mechanical ventilation; provokes more prematurity of neonates; and increases maternal and fetal mortality because of pregnancy. Furthermore, especially for mRNA vaccines, there is now sufficient evidence that providing the vaccines does not cause supplementary side effects for the pregnancy of the fetus. As an extra advantage, there is good evidence the neonate obtains transplacental antibodies at birth, protecting the neonate from neonatal COVID-19 infection. Finally, there is no reason to believe that vaccination in first or third trimester of pregnancy would cause more harm than in the second trimester (when maternal vaccines are usually provided), nor is there a contraindication to provide the vaccine when there is a desire for pregnancy or in the postpartum.

As a result, we firmly advise every pregnant women to be vaccinated with priority at any given time of pregnancy, preferably by mRNA-based vaccines as the side-effect signals are lowest and the experience in pregnancy the highest with these vaccines. In addition, before and after pregnancy, it is safe to provide the vaccination against Sars-CoV-2.

The ethics guidance involved in the current recommendations can be consulted at https://www.sciencedirect.com/science/article/pii/S0264410X19300453 (accessed on 2 May 2021).

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
