# Peer review of "ISIDOG Consensus Guidelines on COVID-19 Vaccination for Women before, during and after Pregnancy"

_jcm, 2021, doi:10.3390/jcm10132902_

Round 1

Reviewer 1 Report

This is a very important topic and it is good to see some work to develop consensus around the science and policy positions. However, there are a number of ways that this piece can be strengthened. The first is that it is not entirely clear who the audience is for this piece - is this meant for health providers, policymakers, women considering vaccination? It would help to clarify who these statements guidelines are intended for. The second note is that some of the conclusions are merely summarizing the nature of the evidence or country positions and others take a normative stance on prioritization and use in pregnancy. It might be helpful to differentiate the bits summarizing the empirical information from the normative guidance. Along those lines, it would be good to conclude with the set of normative guidance that the end, in light of all the relevant information presented.

For more specific feedback on sections of the document, see below:

On page 3, the authors note  "the lowered immunity causes more sensitivity for infection...". It is now widely documented the maternal immune system is not suppressed during pregnancy, although there are some dynamic changes to the immune system across pregnancy (see Mor et al: https://www.nature.com/articles/nri.2017.64). I'd encourage the authors to adopt language that does not reinforce the oversimplistic and incorrect notion of pregnancy as being immunosupressive, though they could make note of changes in the immune system during pregnancy, alongside other physiological changes that could impact progression of disease.

On page 4, as stated, Conclusion 1 seems to contradict Conclusion 2. Is it meant to convey the types of symptoms are the same (even if severity differs?). Please clarify.

On page 5 under Neonatal outcome, "termination of pregnancy" seems to be inappropriately used to refer to early delivery (via c-section). Would not want readers to conflate this with late term abortion - as I am not aware of any clinicians terminating pregnancies to manage COVID-19.

On page 5, the authors say "In addition the fetus should not be exposed to harmful or unknown substances." This feels overly simplistic in a number of ways. First, grouping harmful and unknown is problematic - as there are different considerations at play for a known teratogen or substance with known risk, versus the vast category of unknowns (where there may or may not be biological plausibility for harm). Second, if we never explored "unknown" substances, we would perpetuate the harmful, near categorical exclusion of pregnant people from research - and all substances would remain unknown in pregnancy indefinitely... as the authors note below. Third, there have been many normative guidances that speak to the need to assess risk and benefit across the maternal fetal dyad, and many interventions where, even when there may be some known fetal risk, the benefits to the woman and/or fetus/child outweigh the harms (DTG - and all ARVs for the matter, being prime examples). So I would recommend the authors amend this to not perpetuate the precautionary principle (which they allude to by the "safety" principle) and include more updated approaches to consider risks and benefits for both entities.

I've noticed that the authors discuss later on the emerging findings about rare clotting/thrombotic events associated with some of the vaccines, but it may be worthwhile lumping that with the section on Adenovirus vectored products, and perhaps note that some authorities have noted a preference for mRNA vaccines where available in pregnancy - and decisions to use the adenovectored vaccines should do risk benefit based on best available data on potential thromboembolitic AE and in light of exposure risk to COVID based on the epidemiological situation.

This sentence needs to be revised to be more clear: "Strikingly, 94 and 100% of the anaphylactic reactions in both vaccinated groups occurred in women, 77 and 88% having had allergic reactions before, of which 25 and 34% wee anaphylactic, respectively." Is this saying that there were 94 cases of anaphylaxis captured in VAERS, and all were women? Is there not also trial data that can be referenced? They may also want to update the V-SAFE figures to reflect the latest publication (https://www.nejm.org/doi/full/10.1056/NEJMoa2104983) 

Conclusion 6 notes that, "women in general, and particularly during pregnancy, are more vulnerable to anaphylactic reactions and thrombo-embolic events" - while I'm aware that pregnancy carries its own risks for thrombosis, it is unclear that pregnant women would be more likely to experience the specific types of thromboembolic events that seem to be associated with the vaccines. I agree that more surveillance needs to be done (although, with the rarity of these events, depending on the numbers of pregnant women vaccinated, it may be extremely difficult to assess association or causality as an AEFI). And I'm also unaware of pregnant women being more likely to experience anaphylaxis - or is this just an extrapolation from the fact that women more frequently experience this response than men (even if relatively rare)? And given that the findings suggest those who experience anaphylaxis tend to have a history of severe allergic reactions, it seems worth contextualizing this in discussion of risks/benefits for individuals who have this specific history. I'd suggest the authors tease out the conclusions as they relate to anaphylaxis and thrombosis (specifically VITT/TTS).

On page six, the authors state, "The American College of Obstetrics and Gynecology, the Canadian Society of Obstetrics and Gynaecologists, the German Society for Gynecology and Obstetrics and the Flemish Society of Obstetricians and Gynecologists (VVOG) all favor vaccination for all pregnant women and claim that 'COVID-19 vaccines should not be withheld from pregnant individuals, nor from breastfeeding women.' " Saying that these societies "favor" vaccination may be going a bit further in characterizing the position that pregnant women should have the option to be vaccinated. I suggest amending to say that all these societies "favor pregnant women having the option to be vaccinated and state that 'COVID-19 vaccines should not be withheld..." Similar note to have a bit more nuance characterizing the UK's guidance (as far as I know, as of this writing, only Israel has issued a full recommendation in favor of vaccinating in pregnancy, whereas the others listed defer to a more individual choice/assessment of benefits and risks) - though the inclusion of pregnant women as a priority group in vaccine rollout plans carries an implicit recommendation.

The WHO position is permissive, and specifically calls out pregnant women at high risk for whom the benefits of vaccination likely outweigh risk - though it is worth noting there are some differences in the phrasing between the WHO SAGE Interim Recommendations for different products (https://www.who.int/publications/i/item/WHO-2019-nCoV-vaccines-SAGE_recommendation-BNT162b2-2021.1 ... https://www.who.int/publications/i/item/WHO-2019-nCoV-vaccines-SAGE_recommendation-AZD1222-2021.1 - and all to be updated soon aside from the AstraZeneca one which was recently amended) - and some of the other communications materials/FAQ pages (https://www.who.int/news-room/feature-stories/detail/who-can-take-the-pfizer-biontech-covid-19--vaccine). All are permissive and call for an assessment of risk and benefit. It is therefore not that different from the Netherlands position.

In light of the above, Conclusion 7 should be amended to say that most countries (in the EU - note there are different positions in S. American and Asian countries) favor pregnant women having the opportunity to be vaccinated, especially when there is a favorable risk benefit ratio.

There is some awkward phrasing at the start of section 3.4. This section could benefit from more clearly differentiating what we know about different vaccine platforms in pregnancy versus what we know about specific COVID-19 vaccine products. It may also be a stretch to say that "J&J/Janssen’s Ebola vaccine, based on a similar structure, has shown to be very safe in pregnancy." There have been some studies with are reassuring about safety in pregnancy, but the sample size of those studies the scale of COVID-19 vaccine distribution means that there may be much rarer AEFI that we are able to detect in the context of COVID-19 vaccine pharmacovigilence that would be impossible to observe in a trial (as is the case for the general population). This needs to be contextualized.

Conclusion 8 may also want to caveat deference to mRNA vaccines "where available" recognizing that there may not always be this option in certain contexts.

I would push the authors on conclusion 9 to avoid first trimester. For women with high exposure risk, it may not be wise to delay vaccination given risk of severe disease and death. This section also only talks about passive antibody transfer as it relates to timing, without accounting for timing for direct protection of the mother. That should be addressed as it is relevant to risk benefit assessments across gestation.

On page 8 it is a bit strange to use a term with a different acronym for "fertility enhancing procedures (ART)" - the right longhand for that would be assisted reproductive technologies.

Also on page 8, this sentence needs references: "Other arguments are that SARS-CoV-2 antibodies are passed over by breast milk and that most neonatal infections are passed on by caregivers who are not the parents. " And is the latter specific to COVID-19 and the European setting, or is that a general statement (needs to be clarified and substantiated)

For Conclusion 12, "encouraged" is probably a better word than "motivated" for what the authors are trying to convey.

The authors may want to include some additional references to ethics guidance relevant to the positions they take, such as:

https://www.sciencedirect.com/science/article/pii/S0264410X19300453

Author Response

Comments and Suggestions for Authors

This is a very important topic and it is good to see some work to develop consensus around the science and policy positions. However, there are a number of ways that this piece can be strengthened. The first is that it is not entirely clear who the audience is for this piece - is this meant for health providers, policymakers, women considering vaccination? As stated in the abstract, under the heading ‘Methods’ It would help to clarify who these statements guidelines are intended for. In the Methods section of the text we added the following (in yellow):’This work was done by a team of leading expert obstetricians to guide fellow health care workers in the management of the pregnant women at risk for by Sars-CoV-2 infection’. Of course we hope that policy makers in countries were no national guidelines are in place also will pick up the message. The second note is that some of the conclusions are merely summarizing the nature of the evidence or country positions and others take a normative stance on prioritization and use in pregnancy. It might be helpful to differentiate the bits summarizing the empirical information from the normative guidance. Along those lines, it would be good to conclude with the set of normative guidance that the end, in light of all the relevant information presented. Your remark is entiery correct and well-taken, thank you. We added in the end the following chapter:Data are accumulating that Covid-19 infection has a higher risk to cause severe disease, requiring more admissions to intensive care units, increase the need for mechanical ventilation, provokes more prematurity of the neonates, and increase maternal and fetal mortality because of pregnancy. Furthermore, especially for mRNA vaccines, there is now sufficient evidence that the providing the vaccines does not cause supplementary side effects for the pregnancy of the fetus. As an extra advantage, there is good evidence the neonate will get transplacental antibodies at birth, protecting him from neonatal Covid-19 infection. Finally there is no reason to believe that vaccination in first or third trimester of pregnancy would cause more harm than in the second trimester (when maternal vaccines are usually provided), nor is there a contra-indication to provide the vaccine when there is a desire for pregnancy or in the postpartum. 

As a result we firmly advice every pregnant women to be vaccinated with priority at any given time of pregnancy, preferably by mRNA based vaccines as the side-effect signals are lowest and experience in pregnancy the highest with these vaccines. Also before and after pregnancy it is safe to provide the vaccination against Sars-CoV-2.”

For more specific feedback on sections of the document, see below:

On page 3, the authors note  "the lowered immunity causes more sensitivity for infection...". It is now widely documented the maternal immune system is not suppressed during pregnancy, although there are some dynamic changes to the immune system across pregnancy (see Mor et al: https://www.nature.com/articles/nri.2017.64). I'd encourage the authors to adopt language that does not reinforce the oversimplistic and incorrect notion of pregnancy as being immunosupressive, though they could make note of changes in the immune system during pregnancy, alongside other physiological changes that could impact progression of disease.We adapted the text as follows:  ‘…. the altered immune system, alongside other physiologic changes in pregnancy, causes more sensitivity for infection Also the reference  of Mor et al you kindly provided was introduced.

On page 4, as stated, Conclusion 1 seems to contradict Conclusion 2. Is it meant to convey the types of symptoms are the same (even if severity differs?). Please clarify. I see your point. We adjusted as follows:  Conclusion 1. Infection risk and symptoms of uncomplicated Sars-Cov-2 are not substantially different in pregnant versus non-pregnant women (grade A). 

Conclusion 2. In some cases, the course and severity of Sars-Cov-2 infection is worsened by pregnancy, especially in combination with additional risk factors such as increased maternal age, preeclampsia, obesity, diabetes and hypertension (Grade A).  Therefore, pregnancy should be considered as a priority for vaccination.

On page 5 under Neonatal outcome, "termination of pregnancy" seems to be inappropriately used to refer to early delivery (via c-section). Would not want readers to conflate this with late term abortion - as I am not aware of any clinicians terminating pregnancies to manage COVID-19. Agree! we adjusted as follows:’… mostly because obstetricians choose for induction of labor or primary C-section in order to….’

On page 5, the authors say "In addition the fetus should not be exposed to harmful or unknown substances." This feels overly simplistic in a number of ways. First, grouping harmful and unknown is problematic - as there are different considerations at play for a known teratogen or substance with known risk, versus the vast category of unknowns (where there may or may not be biological plausibility for harm). Second, if we never explored "unknown" substances, we would perpetuate the harmful, near categorical exclusion of pregnant people from research - and all substances would remain unknown in pregnancy indefinitely... as the authors note below. Third, there have been many normative guidances that speak to the need to assess risk and benefit across the maternal fetal dyad, and many interventions where, even when there may be some known fetal risk, the benefits to the woman and/or fetus/child outweigh the harms (DTG - and all ARVs for the matter, being prime examples). So I would recommend the authors amend this to not perpetuate the precautionary principle (which they allude to by the "safety" principle) and include more updated approaches to consider risks and benefits for both entities. We fully agree.  Only we wanted to give the people who are against vaccination with a new vaccin ‘a voice’. We skipped the sentence ‘In addition the fetus…..’

I've noticed that the authors discuss later on the emerging findings about rare clotting/thrombotic events associated with some of the vaccines, but it may be worthwhile lumping that with the section on Adenovirus vectored products, and perhaps note that some authorities have noted a preference for mRNA vaccines where available in pregnancy - and decisions to use the adenovectored vaccines should do risk benefit based on best available data on potential thromboembolitic AE and in light of exposure risk to COVID based on the epidemiological situation. Thank you for the eloquent remark which we implemented by adding to the adenovector vaccines chapter the following: ‘Still as rare side effects of venous sinus thrombosis of the brain, and splanchnic thromboses only seem to occur in adenovectored vaccines, any further decisions to use these vaccines in pregnancy should do risk benefit based on best available data on po-tential thromboembolitic AE and in light of exposure risk to COVID based on the epi-demiological situation.’

 This sentence needs to be revised to be more clear: "Strikingly, 94 and 100% of the anaphylactic reactions in both vaccinated groups occurred in women, 77 and 88% having had allergic reactions before, of which 25 and 34% wee anaphylactic, respectively." Is this saying that there were 94 cases of anaphylaxis captured in VAERS, and all were women? Is there not also trial data that can be referenced? They may also want to update the V-SAFE figures to reflect the latest publication Thank you, we corrected as follows: Strikingly, 94 and 100% of the anaphylactic reactions in both vaccinated groups, with Pfizer-BioNTech and Moderna, respectively, occurred in women. Of them, respectively 77 and 88% had allergic reactions before, of which 25 and 34% wee anaphylactic,’ and the reference was added:https://www.nejm.org/doi/full/10.1056/NEJMoa2104983

Conclusion 6 notes that, "women in general, and particularly during pregnancy, are more vulnerable to anaphylactic reactions and thrombo-embolic events" - while I'm aware that pregnancy carries its own risks for thrombosis, it is unclear that pregnant women would be more likely to experience the specific types of thromboembolic events that seem to be associated with the vaccines. I agree that more surveillance needs to be done (although, with the rarity of these events, depending on the numbers of pregnant women vaccinated, it may be extremely difficult to assess association or causality as an AEFI). And I'm also unaware of pregnant women being more likely to experience anaphylaxis - or is this just an extrapolation from the fact that women more frequently experience this response than men (even if relatively rare)? And given that the findings suggest those who experience anaphylaxis tend to have a history of severe allergic reactions, it seems worth contextualizing this in discussion of risks/benefits for individuals who have this specific history. I'd suggest the authors tease out the conclusions as they relate to anaphylaxis and thrombosis (specifically VITT/TTS). Agree! We added the following to conclusion 6: ‘It should be clear, however, that there are no signals that the extremely rare side ef-fects linked to the adenovirus vectored vaccines, are more frequent during pregnancy. Also are the anaphylactic reactions rather linked to an history of anaphylaxis that to gestation itself.

On page six, the authors state, "The American College of Obstetrics and Gynecology, the Canadian Society of Obstetrics and Gynaecologists, the German Society for Gynecology and Obstetrics and the Flemish Society of Obstetricians and Gynecologists (VVOG) all favor vaccination for all pregnant women and claim that 'COVID-19 vaccines should not be withheld from pregnant individuals, nor from breastfeeding women.' " Saying that these societies "favor" vaccination may be going a bit further in characterizing the position that pregnant women should have the option to be vaccinated. I suggest amending to say that all these societies "favor pregnant women having the option to be vaccinated and state that 'COVID-19 vaccines should not be withheld..." Similar note to have a bit more nuance characterizing the UK's guidance (as far as I know, as of this writing, only Israel has issued a full recommendation in favor of vaccinating in pregnancy, whereas the others listed defer to a more individual choice/assessment of benefits and risks) - though the inclusion of pregnant women as a priority group in vaccine rollout plans carries an implicit recommendation. We adapted the text as follows: “The American College of Obstetrics and Gynecology, the Canadian Society of Obstetrics and Gynaecologists and the German Society for Gynecology and Obstetrics all favor pregnant women having the option to be vaccinated and state that "COVID-19 vaccines should not be withheld from pregnant individuals, nor from breastfeeding women." And further on: “Even more pronounced, the Flemish Society of Obstetricians and Gynaecologiscts (VVOG) and the Israeli Society for OB/GYN favor vaccination of pregnant women as a priority group.”  

The WHO position is permissive, and specifically calls out pregnant women at high risk for whom the benefits of vaccination likely outweigh risk - though it is worth noting there are some differences in the phrasing between the WHO SAGE Interim Recommendations for different products (https://www.who.int/publications/i/item/WHO-2019-nCoV-vaccines-SAGE_recommendation-BNT162b2-2021.1 ... https://www.who.int/publications/i/item/WHO-2019-nCoV-vaccines-SAGE_recommendation-AZD1222-2021.1 - and all to be updated soon aside from the AstraZeneca one which was recently amended) - and some of the other communications materials/FAQ pages (https://www.who.int/news-room/feature-stories/detail/who-can-take-the-pfizer-biontech-covid-19--vaccine). All are permissive and call for an assessment of risk and benefit. It is therefore not that different from the Netherlands position. We agree and added the WHO extra reference

In light of the above, Conclusion 7 should be amended to say that most countries (in the EU - note there are different positions in S. American and Asian countries) favor pregnant women having the opportunity to be vaccinated, especially when there is a favorable risk benefit ratio.

We added ‘European and North American’

There is some awkward phrasing at the start of section 3.4. This section could benefit from more clearly differentiating what we know about different vaccine platforms in pregnancy versus what we know about specific COVID-19 vaccine products. It may also be a stretch to say that "J&J/Janssen’s Ebola vaccine, based on a similar structure, has shown to be very safe in pregnancy." There have been some studies with are reassuring about safety in pregnancy, but the sample size of those studies the scale of COVID-19 vaccine distribution means that there may be much rarer AEFI that we are able to detect in the context of COVID-19 vaccine pharmacovigilence that would be impossible to observe in a trial (as is the case for the general population). This needs to be contextualized. Entirely correct! We added the following: ‘However, given he small sample size of these studies as compared to the Covid-19vaccines currently applied worldwide, it may well be that the rare throm-bolembolic events such as venous sinus thrombosis of the ebola vaccines are still ‘un-der the radar’.

Conclusion 8 may also want to caveat deference to mRNA vaccines "where available" recognizing that there may not always be this option in certain contexts.

Agree, we added ‘were available’

 I would push the authors on conclusion 9 to avoid first trimester. For women with high exposure risk, it may not be wise to delay vaccination given risk of severe disease and death. This section also only talks about passive antibody transfer as it relates to timing, without accounting for timing for direct protection of the mother. That should be addressed as it is relevant to risk benefit assessments across gestation. I am glad you are on our side also in this one. We adapted as follows: ‘Preferable time for vaccination in pregnancy is the second trimester, but if women are encountered first during first or third trimester there is no reason to refrain from vac-cination, as the benefits for the mother always outweigh a possible Covid-19 infections during pregnancy (Grade B).’

On page 8 it is a bit strange to use a term with a different acronym for "fertility enhancing procedures (ART)" - the right longhand for that would be assisted reproductive technologies.

As both terms are used in different regions, we added the full longhand of ART but left the fertility enhancing procedures as well. Thank you for the attentive remark

 Also on page 8, this sentence needs references: "Other arguments are that SARS-CoV-2 antibodies are passed over by breast milk and that most neonatal infections are passed on by caregivers who are not the parents. " And is the latter specific to COVID-19 and the European setting, or is that a general statement (needs to be clarified and substantiated)

We added 3 references in breast milk protection and a reference of horizontal transmission. The European setting is recognized by adding ‘according to an Italian study(. In summary the sentence translates now as follows: ‘Other arguments are that SARS-CoV-2 antibodies are passed over by breast milk (22-24), especially after vaccination (25) and that most neonatal infections are passed on by caregivers who are not the parents, according to an Italian study (26)’. 4 references were included.

For Conclusion 12, "encouraged" is probably a better word than "motivated" for what the authors are trying to convey.We replaced ‘motivated’ by ‘encouraged’

The authors may want to include some additional references to ethics guidance relevant to the positions they take, such as:

https://www.sciencedirect.com/science/article/pii/S0264410X19300453

The following sentence was added to the paper in the summary section to inform readers about the ethical issues involved. Thank  you for this excellent input!  ‘The ethics guidance involved in the current recommendations can be consulted at https://www.sciencedirect.com/science/article/pii/S0264410X19300453.’

Submission Date

19 April 2021

Date of this review

28 Apr 2021 18:35:23

Reviewer 2 Report

Donders et al., in this "ISIDOG consensus guidelines on COVID19 vaccination for women before, during and after pregnancy" paper have performed an extensive literature search regarding COVID19 effects on pregnancy, COVID vaccine side effects, and the available information regarding COVID vaccination during pregnancy and breastfeeding. The review and conclusions on the vaccination recommendation are of great interest for a priority publication 

Specific comments:  

-31. Neonatal outcome:There is a mistake in the bibliography. Reference 10 corresponds to 9 and vice versa

- 3.5: Timing of vaccination: You do not suggest a recommendation in the case of a patient with a first dose given periconceptionally or early in pregnancy before knowing  she was pregnant. Would you recommend the second dose given at the scheduled time during the first trimester or to postpone it? This information would be of interest.

Author Response

Donders et al., in this "ISIDOG consensus guidelines on COVID19 vaccination for women before, during and after pregnancy" paper have performed an extensive literature search regarding COVID19 effects on pregnancy, COVID vaccine side effects, and the available information regarding COVID vaccination during pregnancy and breastfeeding. The review and conclusions on the vaccination recommendation are of great interest for a priority publication 

Thank you for the kind support

Specific comments:  

-31. Neonatal outcome:There is a mistake in the bibliography. Reference 10 corresponds to 9 and vice versa

Thank you so much for noticing this. We changed the references accordingly

- 3.5: Timing of vaccination: You do not suggest a recommendation in the case of a patient with a first dose given periconceptionally or early in pregnancy before knowing  she was pregnant. Would you recommend the second dose given at the scheduled time during the first trimester or to postpone it? This information would be of interest.

Thank you for this highly important remark. We added the following: ‘As the potential side effects are very rare, and even less likely to occur after a second dose if the first dose went uneventfully, we advise to continue the vaccination schedule according to plan if a person fell pregnant between the first and the second dose of a vector vaccine.’

Submission Date

19 April 2021

Date of this review

07 May 2021 11:52:36

Round 2

Reviewer 1 Report

The paper has definitely improved and the authors were attentive to the comments and revisions requested.

Conclusion 9 is very hard to interpret. The revised, highlighted language seems to imply that the benefits outweigh Covid 19 infection, rather than "the benefits of vaccination to protect against COVID-19 in pregnancy are likely to outweigh risks of vaccination, regardless of trimester", the latter being what I presume the authors are trying to say.

The paper could still use a copy editor to improve the sentence flow/English language and there are still a number of typos (e.g. "sore throat" not "soar" on p.3 and "advise" not "advice" in the conclusion)

If possible to have someone copy edit for both awkward phrasing and typos that would help.

Author Response

The paper has definitely improved and the authors were attentive to the comments and revisions requested. Thank you. We were impressed iof your intense review and thank you for helping us to improve the paper

Conclusion 9 is very hard to interpret. The revised, highlighted language seems to imply that the benefits outweigh Covid 19 infection, rather than "the benefits of vaccination to protect against COVID-19 in pregnancy are likely to outweigh risks of vaccination, regardless of trimester", the latter being what I presume the authors are trying to say. Both prevention of Covid-19 infection and the minimal risks af the vaccine are reasons to accept the vaccine. So we adapted the sentence as follows: "...there is no reason to refrain from vaccination, as the benefits for the mother are likely to outweigh the risks of vaccination or a possible Covid-19 infection during pregnancy, regardless the trimester (Grade B). "

The paper could still use a copy editor to improve the sentence flow/English language and there are still a number of typos (e.g. "sore throat" not "soar" on p.3 and "advise" not "advice" in the conclusion)

The typo's are corrected

If possible to have someone copy edit for both awkward phrasing and typos that would help.

Indeed our mother tongue is not English, explaining these feelings, for which we apologize. Although the text was already copy-edited by a British co-author we would welcome any further improvements on awkward phrasing and typo's.